# Inequality of opportunity in child nutrition in Pakistan

**Shyamkumar Sriram**[1]*, **Lubna Naz**[2]

**1** Department of Rehabilitation and Health Services, University of North Texas, Denton, Texas, United States of America, **2** Department of Economics, Institute of Business Administration, School of Economics and Social Sciences, Karachi, Pakistan

\* shyam.silverhawk@gmail.com

## Abstract

### Introduction

Malnutrition among children is one of the major health challenges in Pakistan. The National Nutritional Survey 2018 revealed that 44% of children are stunted. Different circumstances surrounding a child's birth can lead to inequality of opportunity in early childhood, with significant nutritional inequalities between rural and urban areas. This study aims to identify the drivers of inequality of opportunity in stunting among children under-five years of age in Pakistan.

### Methods

This study used Pakistan Demographic and Health Survey, 2017–18 to identify the factors contributing to inequality of opportunity in child's stunting. The Dissimilarity index (D-index), along with Oaxaca decomposition, and Shapely decomposition were employed to measure and decompose inequality in opportunity in stunting. Regional variations in stunting among children under various circumstances were analyzed using Geographic Information System or GIS.

### Results

The burden of stunting is exceptionally high in Pakistan, with the prevalence in rural areas significantly exceeding that in urban areas from 1990 to 2018. Shapley decomposition of the contributors to inequality in opportunity indicates that maternal education accounted for 24% of total inequality among rural children and 44% among urban children. Water and sanitation contributed 22% to overall inequality in rural areas but only 2% in urban areas, highlighting the critical role of inadequate water and sanitation in rural settings. The wealth index was a predominant contributor to inequality both nationally and in urban areas. Southern regions exhibit a higher prevalence of stunting and a greater proportion of households lacking adequate water and sanitation. Additionally, the concentration of uneducated mothers and stunted children is notably high in Balochistan and Sindh.

**Data Availability Statement:** Third party data was obtained for this study from the DHS Program. This study is based on data obtained from the Pakistan Demographic Health Survey data (PDHS) 2017-18. Data may be requested from the DHS Program after creating an account and submitting

a concept note. More access information can be found on the DHS Program website (https://dhsprogram.com/data/Access-Instructions.cfm). The authors confirm that interested researchers would be able to access these data in the same manner as the authors. The authors also confirm that they had no special access privileges that others would not have.

**Funding:** The author(s) received no specific funding for this work.

**Competing interests:** The authors have declared that no competing interests exist.

## Conclusions

The lack of maternal education, inadequate access to water and sanitation services, and lower socio-economic status are key factors contributing to inequality of opportunity in stunting among children under five in Pakistan. Understanding the critical role of these circumstances can help policymakers address the situation and implement concrete steps to enhance equal opportunities for child health.

## 1. Introduction

Malnutrition in all forms, ranging from stunting to obesity is currently one of the major health challenges worldwide [1]. Stunting is the result of poor nutrition during the prenatal stage and early childhood. Stunted children are less likely to reach their potential height. Globally, 22% were stunted (below the standard height for age) in 2022. Of these 52% live in Asia, and 43% in Africa. Similarly, 6.8% of children are currently suffering from wasting (weight for height) in the world. Amongst 13.6 million (2.1%) children severely wasted, three quarters live in Asia and another 22% live in Africa. Wasting is a life-threatening outcome which results from inadequate nutrient intake. Children suffering from wasting tend to be dangerously thin, have weakened immunity, and face a higher risk of mortality [2].

South Asia has been grappling with a high burden of malnutrition, with 28 million and 62 million children are stunted and wasted, respectively [3]. About one in three children were stunted, one in seven children wasted and one in twenty-two are severely wasted in South Asia. Alarmingly, severe wasting accounts for 7.7 million children in South Asia which is three times as many as in Sub-Saharan Africa [4].

Moreover, huge disparities exist in childhood nutrition across geographic locations and household wealth. In the past few decades, the urban population has grown at an unprecedented pace, with 55% of the total global population living in urban areas. By 2050, 68% of the world population is expected to be in urban areas, with 90% of this increase projected to take place in Asia and Africa [5]. Place of residence determines the availability of opportunities, especially in developing countries where significant disparities prevail [6]. Rural residents face several challenges, such as limited access to health services, food insecurity, and a lack of clean water, sanitation, and hygiene [7]. The risk of poor cognitive development has been found to be higher among rural children compared to their urban counterparts [8]. A study showed that nearly 43% of children under five years of age from low- and middle-income countries (LMIC) in 2010 were at risk of poor development, with the highest proportion of affected children in South Asia and Sub-Saharan Africa.

Early childhood exposure to health inequalities can have lasting adverse effects, transmitting to the next generation, leading to the never-ending cycle of poverty and inequality The literature divides inequality of opportunity into legitimate and illegitimate sources. Legitimate sources refer to the factors which are under the control of individuals and determined by individual behavior. In contrast, illegitimate sources are associated with circumstances not under the control of individuals, such as ethnicity, the characteristics of the place where an individual is born or grows up, parental background, or early childhood conditions [9]. Inequality of opportunity in child health is influenced by the social and economic circumstances and environment into which children are born.

Children can neither choose their circumstances nor the environment into which they are raised [10]. The mental and physical development of child health occurs in the early years of

life. An inequality of opportunity in children's nutritional outcomes was found to be higher in rural areas compared to urban, and family socioeconomic status contributed to these disparities in China. Although, the health status of Chinese children improved over the years, inequality between privileged and underprivileged children had widened [11].

Furthermore, factors such as limited access to clean water and sanitation and region of residence are indicators of malnutrition among children [12–14]. Deprivation of clean water and sanitation and unhygienic practices multiply the risk of malnutrition. Differences in residence are another principal factor that impacts child health and nutrition [15]. emphasized location as a major factor of inequality in child health and found larger differences in the nutritional status of urban and rural children. Moreover, the study highlighted correlation between poor child growth and inadequate WASH (Water, Sanitation and Hygiene).

Children exposed to inadequate sanitation and unhygienic practices have a higher chance of contracting various diseases, which can lead to malnutrition. A study by [16] found differences in sanitary conditions among households in Nepal, India, Cambodia and East Timor, with the presence of flushing toilets identified as a vital contributor to inequality. Their findings indicate the importance of sanitation facilities in reducing malnutrition. Moreover, the inequality in child health persists in countries with high levels of economic disparities. A study by [17] found that child health varies among and within countries, and children from the least advantaged families have severe nutritional outcomes. Addressing the causes behind this inequality is essential to improve well-being for all children.

Socio-economic status and maternal education have been established as significant contributors to health inequality among children [18,19]. Child malnutrition is usually an outcome of poverty and low family income. Poverty intensifies the risk of malnutrition. Poor children are likely to be affected by limited financial resources that restrict their nutritional needs, which can lead to health issues like stunting and wasting [20]. The analysis also found that the wealth status is associated with child height positively, with children from richest quantiles being, on average, taller than children from less affluent quantiles [13].

Moreover, mothers with limited education and a lack of knowledge face constraints in decision-making regarding their children's nutrition. A study showed that a mother's education and wealth status are highly significant factors for child nutrition status in Sub-Saharan Africa. These factors are beyond the control of children under five years of age, and these initial years of their life are most important for both physical and mental development [20]. Addressing health differences at the childhood level needs crucial attention to improve stunting and wasting because the negative effects of malnutrition are persistent and have long-term impacts.

In Pakistan, 40% of children under five years of age are stunted, while 17.7% suffer from wasting. The prevalence of stunting and wasting is more prominent in rural areas than in urban areas. Among the children who are suffering from stunting, 43% belong to rural areas, whereas 34.8% live in urban areas. Moreover, 18.6% of rural children are afflicted with wasting, while in urban areas, this figure is 16% [21]. According to PDHS (2017–18), stunting and wasting declined from 45% to 38% and 11% to 7%, respectively, between 2013–14 and 2017–18. Stunting is high among children in the lowest wealth quintile, which is 57%, compared with the highest wealth quintile, which is half of what poor children experience. Previous evidence has established the presence of massive disparities in nutritional status among rural and urban children in lower wealth quintiles [22–24].

Pakistan's progress in child nutrition and health has been substantially slower than that of other South Asian countries. Prior research by [25] in the context of Pakistan shows families with low socio-economic status and lower maternal education restrict the role of the mother in household decision-making. This limits the capacity of mothers to make the right decision about the health and nutritional requirements of their children, resulting in poor nutritional

status, including stunting and wasting. Stunting and wasting were highest amongst children belonging to the poorest quintile and whose mothers had no education [21]. Moreover, rural women work for low wages due to unavailability of work for the full year in agriculture; this has an impact on the health of their children, as mothers don't have enough money to spend on their children [26].

The existing research on child nutrition in Pakistan shows that education, geographical, and socioeconomic differentials are critical contributors to increasing inequalities in child health. Additionally, breastfeeding has been identified as a contributing factor to the nutritional status of children. Pakistan has ranks some of the lowest rates for early breastfeeding initiation and exclusive breastfeeding [23,27]. However, the studies revealed an absence of significant gender-related differences in child nutritional status in Pakistan, which contrasts with the widely held perceptions of treating male children favorably than female children in Pakistan. These studies have highlighted the strong effects of inadequate sanitation facilities on child nutritional status [28]. Children from poor households often face challenges in accessing sufficient food and healthcare services, making them more susceptible to growth problems. A study by [26] assessed the contributors to child health inequalities in Pakistan, revealing that sanitation, rural residence, mother's education, and wealth status are the leading contributors to these inequalities. These inequalities are further compounded by regional disparities in socioeconomic development. Several regions face challenges related to water and sanitation facilities. The absence of piped water in homes complicates hygienic practices for people. To access water, individuals in rural areas must travel long distances from their residences to the water source. This situation not only makes daily life unhygienic but also increases the risk of diseases among children [29,30].

Children from poor households often face challenges in accessing sufficient food and healthcare services, making them more susceptible to growth problems. A study by [26] assessed the contributors to child health inequalities in Pakistan, revealing that sanitation, rural residence, mother's education, and wealth status are the leading contributors of inequalities. These inequalities are further compounded by regional disparities in socio-economic development [29,30].

While national-level studies on child stunting are abundant, there is limited research focusing on the sub-national or provincial disparities in Pakistan. Many studies tend to aggregate the data at the national level, masking regional inequalities that may have distinct drivers and implications. Moreover, the majority of previous stunting studies focus on inequality in outcomes (such as the prevalence of stunting), but relatively few apply the inequality of opportunity framework for deeper understanding of structural inequalities, which distinguishes between factors children cannot control (e.g., parental education, socioeconomic background) and those they can (e.g., individual efforts). Further, there is a dearth of literature in unravelling the contribution of various factors in overall inequality in stunting via decomposition techniques. Additionally, very few studies in Pakistan have explored the association between access to water and sanitation and stunting. Hence, this study was undertaken to bridge the gap in the previous research using a more rigorous and nuanced approach.

## 2. Methodology

### 2.1 Dataset and sample

This study extracted a sample of 4,132 of children from the Pakistan Demographic Health Survey (PDHS) 2017–18, which is a nationally representative survey. The dataset provides comprehensive information on a wide range of indicators in the areas of population, health, and nutrition. The National Institute of Population Studies (NIPS) implemented the survey in

coordination with the Ministry of National Health Services, Regulations and Coordination Pakistan. Data collection took place from November 22, 2017, to April 30, 2018. The sampling frame comprises a complete list of enumeration blocks (EBs) created for the Pakistan Population and Housing Census 2017 [31]. The PDHS employs a two-stage stratified random sampling technique, where 580 enumeration areas (EAs) are selected as the primary sampling units in the first stage, followed by the random selection of 28 households within each EA in the second stage. Information was collected from a random sample of 16,240 households, whereas anthropometric information was gathered for only one-third of the households. Data on height, weight, and age were taken for children aged 0–59 months. This study calculated height-for-age, weight-for-height, and weight-for-age scores. The selected sample represents urban and rural areas and is designed to provide estimates at the national and provincial levels.

Each of these indices provides valuable insights about the growth of children in the early years of life, which helps in the assessment of nutritional status.

## 2.2 Ethics statement

This study is based on secondary data that was obtained from the DHS program website. Prior permission was obtained from ICF to download the data and use it in the study. https://dhsprogram.com/data/available-datasets.cfm. The data is in the open domain. Anyone can access the data and replicate the results with prior permission of the DHS program-ICF.

**2.2.1 Variables.** Stunting was used as the outcome variable in the study. Children whose height-for-age Z-score is below two standard deviations (-2 SD) from the median of the reference population are considered short for their age (stunted) and children who are below three standard deviations (-3 SD) are considered severely stunted. In the 1960s and early 1970s, a large group of children in the United States was weighted and measured for the reference population; this group is called NCHS: CDC: WHO reference. Recently, the World Health Organization has given a new reference population of children from six countries (Brazil, United States, Norway, Ghana, Oman, and India). The nutrition and health of these children were closely monitored; they were exclusively breastfed and completed all necessary vaccinations.

To assess the decomposition of inequality of opportunity in child health, several socio-demographic factors were chosen following the literature. See details in Table 1.

**2.2.2 Theoretical framework.** This study is based on the analytical framework of inequality of opportunity (IOP) in child health, which provides an overview of how circumstances beyond the control of children can affect their health outcomes. This study follows the framework proposed by [9], which posits that inequality of outcomes is ethically acceptable when results from differences in individuals' efforts, while the inequality arising from "uncontrollable circumstances" is morally unacceptable. The latter part of the Roemer argument refers to the inequality of opportunity and it is driven by factors such as parental education, place of residence, socioeconomic background, and access to improved facilities. Moreover, genetic and natural factors affect the height-for-age (stunting) and weight-for-height (wasting) among children. Genetic factors are inherited from biological parents, while environmental factors can be attributed to external conditions. Children have no control over these factors; they are largely dependent on the resources provided to them.

## 2.3 Empirical methods

**2.3.1 Dissimilarity index and human opportunity index.** The analysis of this study is built on the existing methodologies used to measure inequalities in opportunity in child health. We employed the Dissimilarity Index (D-index) to quantify inequality of opportunity at the urban and rural levels. Previous studies have used the D-index to measure the dissimilarity of

**Table 1.**

| Variables Names | Measurement Level (categorical/binary/continuous) | RATIONALE | Reference of studies which have used similar variables |
|---|---|---|---|
| Coverage of basic vaccinations | Binary | Immunization protects children from infectious diseases that can help prevent the malnourishment process. | [14,32] |
| Mother's Education | Categorical | Mother's education impacts child health outcomes through better knowledge of nutrition, hygiene, and healthcare access. | [16,20,33] |
| Mother's currently working | Binary | Mother's job is likely to have ambiguous effect on the child nutritional status, as working mothers have less time to be spent on childcare, however, mother's income may contribute to higher expenditure on child healthcare and wellbeing. | [8,26] |
| Household Size | Discrete | Larger household sizes may dilute resources (e.g., food, healthcare), increasing the risk of stunting among children. | [11,20,34] |
| Birth Order | Categorical | Higher birth order is associated with resource constraints in the household, leading to poorer nutritional outcomes for children born later. | [35–37] |
| Child Sex | Binary | Sex-based biases exist in Asian culture, leading to gendered differences in stunting, with girls being more likely to be stunted than boys | [6,38] |
| Wealth Index | Categorical | Socio-economic status is a primary driver of inequality in access to food, healthcare, and sanitation, which influences stunting. | [33,39] |
| Water and Sanitation | Binary | Improved water and sanitation access lowers the risk of infections and diarrhea, contributing to better growth outcomes. | [40–42] |
| Child Breastfeeding | Categorical | Exclusive breastfeeding for the first 6 months is associated with better nutrition and reduced stunting | [43,44] |
| Region | Categorical | Regional disparities in governance, infrastructure, poverty levels, and public health service delivery influence stunting due to variations in | [13,19,45] |

[1] Guide to Demographic Health Survey (DHS) Statistics.

access rates for a given health service, such as immunization, prenatal care, cancer screening, and access to treatments for chronic conditions. These studies compare access rates for groups defined by circumstance characteristics (such as gender, location, parental education, and so forth) with the average access rate for the same service for the population [46]. The process of calculating the D-index involves three steps. In the first step, we applied logistic regression to determine the likelihood of each child having access to opportunities conditional on circumstances (x1, x2, x3, ..., xm).

$$ln\left(\frac{Pr(I = 1|x_1, x_2, \ldots\ldots\ldots\ldots, x_m)}{1 - Pr(I = 1|x_1, x_2, \ldots\ldots\ldots, x_m)}\right) = \sum_{k=1}^{m} h_k(x_k) \tag{1}$$

In the second step, we used the coefficients drawn in the first step to predict the probability of the access of opportunity for each child in the sample.

$$\widehat{p} = \frac{exp\left(\beta_0 + \sum_{k=1}^{m} \widehat{X_{ki}\beta_{ki}}\right)}{1 - exp\left(\beta_0 + \sum_{k=1}^{m} \widehat{X_{ki}\beta_{ki}}\right)} \tag{2}$$

The final step draws on the probability of the access to opportunity for the overall population estimated and then used as an input to determine the D-Index.

$$\widehat{p} = \sum_{i=1}^{n} w_i \widehat{p}_i \tag{3}$$

$$D = \frac{1}{2p} \sum_{i=1}^{n} w_i \lceil \widehat{p_i - p} \rceil \tag{4}$$

Where $\widehat{p}_i$ is the predicted probability of an outcome i, $\bar{p}$ is the average probability for the population, and $w_i$ is the sample weight. The D-index ranges from 0 to 1 (or 0 to 100 in percentage terms), and in a situation of perfect equality of opportunity, the D-index will be zero. The index is linked with the coverage when there is complete equality, D = 0 and the value of D decreases when inequality increases ($0 < D \leq 1$) [47].

To measure the overall accessibility rate (prevalence) of opportunities, we employed the Human Opportunity Index (HOI), which quantifies how fairly the opportunities are distributed across population groups. The HOI was calculated by multiplying the product of the average access rate with the D−index.

$$HOI = \underline{p}(1 - D) \tag{5}$$

HOI depicts the distribution of opportunities regardless of the child's circumstances; see Eq 5.

**2.3.2 Shapely decomposition.** This study used the decomposition of inequality of opportunity in child health given by Shapely Decomposition [48] to assess the role of various factors to the inequality; see Table 1 for a list of factors. This method decomposes a given arbitrary function f (.) into its arguments X1, X2, X3. Where f (.) determines the underlying model and represents the level of inequality in population. It shows the marginal contribution of each circumstance and group adds up exactly to the total inequality [49]. Shapley Decomposition is an exact decomposition (and additive) method that has symmetry with respect to the order of the arguments, meaning the contribution attributed to a specific factor remains consistent regardless of the arrangement of the factors. To measure the effect of circumstances, such as maternal education, wealth status, and WASH, in Pakistan, we used the following equation:

$$D_A = \sum SC N \setminus \{A\} \frac{|s|!(n - |s| - 1)!}{n!} [D(S \cup \{A\}) - D(S)] \tag{6}$$

The methodology has been widely used for assessing and decomposing inequality in child health [13,14,20]. [18] applied Shapley Decomposition to understand the impact of numerous early-life circumstances on inequalities in health outcomes, while a similar technique has been used to assess the inequality of a child's nutritional opportunity attributed to water and sanitation [15]. Furthermore, [50] employed Shapley Decomposition to elucidate inequalities in health-related quality of life.

### 2.3.3 Oaxaca decomposition

This study employed the Oaxaca-Blinder decomposition to identify factors causing inequality [51,52]. The decomposition breaks down the differences in an outcome variable between two groups: explained and unexplained parts. The explained part shows the average differences in the predicted probabilities of the least and most advantaged groups, whereas the unexplained part reflects the differences that are not considered in this study.

$$\Delta \underline{Y} = (\underline{X}_A - \underline{X}_B)'\widehat{\beta}_R \underbrace{\phantom{x}} + \underline{X}'_A(\widehat{\beta}_A - \widehat{\beta}_R) \underbrace{\phantom{x}} + \underline{X}'_B(\widehat{\beta}_R - \widehat{\beta}_B) \underbrace{\phantom{x}} \tag{7}$$

*Explained unexplained A     unexplained B* $\underbrace{\phantom{x}}$

*unexplained*

Consider a variable Y, which is the outcome variable showing inequality of opportunity in malnutrition, and we consider the inequality in Group A (rural) and Group B (urban) samples.

The variable Y is explained by a vector of x determinants (for each group) in the linear regression model as follows:

$$IoP^R_{Mean} = \beta^R_{Mean} + \sum\nolimits_{j=1}^{J} . \beta^R_j X^R_{j\ mean} \tag{8}$$

$$IoP^U_{Mean} = \beta^U_{Mean} + \sum\nolimits_{j=1}^{J} . \beta^U_j X^U_{j\ mean} \tag{9}$$

The superscripts R and U represent rural and urban, respectively, while X shows a set of J measured predictors (see Table 1). Inequality of Opportunity is donated by IoP and $IoP_{mean}$, show the mean value of outcome variable., represents the coefficients that show the association between predictors included in $X_j$ and IoP. The difference in the above equation can be expressed as:

$$IoP^R_{Mean} - IoP^U_{Mean} = (\beta^R_0 - \beta^U_0) \tag{10}$$

The Linear Oaxaca-Blinder Decomposition equation can be expressed after applying the simple algebraic manipulation to the equation, as follows:

$$IoP^R_{Mean} - IoP^U_{Mean} = \left[\sum\nolimits_{j=1}^{J}(X^R_{j\ mean} - X^U_{j\ mean})\beta^R_j\right] + \left[(\beta^R_0 - \beta^U_0) + \sum\nolimits_{j=1}^{J}(\beta^R_j - \beta^U_j)X^U_j\_mean\right] \tag{11}$$

The first term on the left-hand represents the "explained" portion that shows the inequality that exists between rural and urban areas caused by a wide array of factors. The second term shows the "unexplained" part, indicating disparities in the coefficients and mean values of the factors.

## 3. Results

### 3.1 Descriptive statistics

Table 2 shows the descriptive statistics of the variables used in the study. The data indicates that 70% of children did not receive the full vaccination. About 49% of mothers have no formal education, 15% have primary qualifications, and 36% have middle, secondary or higher education. A significant majority of mothers (87%) were unemployed, while only 13% were employed. Birth order statistics showed that 25% of children were the firstborn, followed by 38% and 22% as the second and third born, respectively. The wealth index showed a significant number of poor households (42%), 20% were in the middle rank and 38% were rich. One-fourth of households (24%) did not have adequate water and sanitation facilities, while 76% had the facility of water and sanitation. The average size of the household was 9, and 93% of mothers reported exclusive breastfeeding. The correlation coefficients of stunting with mother's education, wealth index, and WASH were—0.23,– 0.22,– 0.18, respectively.

### 3.2 Trends in stunting

Fig 1 shows the trend in stunting for years 1990–91, 2012–13, and 2017–18 using Pakistan Demographic Health Surveys (PDHS) data. The figure indicates regional disparity, with children living in rural areas being more stunted than those living in urban areas. The trend shows a decline in stunting over the last three decades at both urban and rural levels.

### 3.3 Spatial relationship

Figs 2–4 show the spatial relationship of stunting with access to water and sanitation services, education of the mothers, and wealth index, using data from the PDHS (2017–18). Fig 2 shows

**Table 2. Summary statistics 95% conf. interval.**

|  | Proportion | Mean | Lower | Upper |
|---|---|---|---|---|
| Coverage of full vaccinations |  |  |  |  |
| No | 0.70 |  | 0.69 | 0.72 |
| Yes | 0.30 |  | 0.28 | 0.31 |
| Mother's education |  |  |  |  |
| No education | 0.49 |  | 0.47 | 0.50 |
| Primary | 0.15 |  | 0.14 | 0.17 |
| Secondary or higher | 0.36 |  | 0.34 | 0.38 |
| Mother currently working |  |  |  |  |
| No | 0.87 |  | 0.86 | 0.88 |
| Yes | 0.13 |  | 0.12 | 0.14 |
| Birth order number |  |  |  |  |
| 1 | 0.25 |  | 0.24 | 0.27 |
| 2–3 | 0.38 |  | 0.37 | 0.40 |
| 4–5 | 0.22 |  | 0.21 | 0.24 |
| 6+ | 0.14 |  | 0.13 | 0.15 |
| Child Sex |  |  |  |  |
| Male | 0.51 |  | 0.49 | 0.52 |
| Female | 0.49 |  | 0.48 | 0.51 |
| Wealth index |  |  |  |  |
| Poor | 0.42 |  | 0.40 | 0.43 |
| Middle | 0.20 |  | 0.19 | 0.21 |
| Rich | 0.38 |  | 0.37 | 0.40 |
| Water and sanitation |  |  |  |  |
| Inadequate access | 0.24 |  | 0.22 | 0.25 |
| Adequate access | 0.76 |  | 0.75 | 0.78 |
| Household size |  | 9.10 | 9.01 | 9.19 |
| Child breastfeeding |  | 93.67 | 93.65 | 93.69 |

|  | Pairwise correlation | | |
|---|---|---|---|
|  | Mother's education | Wealth index | Water and sanitation |
| Stunting children | -0.23 | -0.22 | -0.18 |

that stunting is more prevalent in areas with inadequate access to water and sanitation facilities, such as the upper portion of Balochistan and Gilgit-Baltistan.

Conversely, areas with better access to water and sanitation have a low prevalence of stunting among children. Fig 3 displays a lower prevalence rate of stunting in regions where mothers were educated. Stunting is higher among children whose mothers lack education, emphasizing a strong relationship between mothers' education and child health. Fig 4 shows a noticeable prevalence of stunting among children from poor families. The spread of stunting is uneven across the country.

### 3.4 Inequality of opportunity and decomposition

Table 3 shows that the value of the adapted D-index is 0.293. Table 3 and Fig 5 indicates that a significant part of inequality of opportunity in child health is linked to circumstances. This can also be regarded as a redistribution of circumstances-driven inequality from non-stunted to stunted children in the pursuit of equality. since not all factors were included in the analysis, the D-index value can be considered a lower bound estimate.

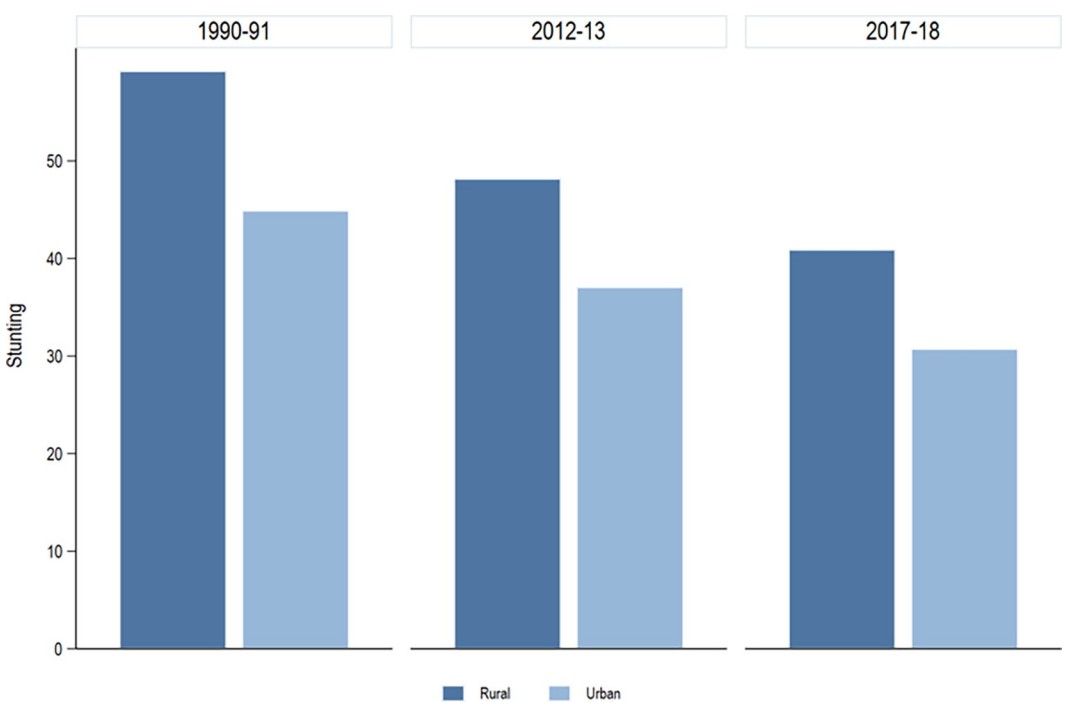

Data source: Pakistan Demographic and Health Survey (PDHS): 1990-91, 2012-13, 2017-18

**Fig 1. Trends in stunting.**

### 3.5 Shapley decomposition

The results of the Shapley Decomposition of the inequality of opportunity in stunting are presented in Table 4. The findings indicate mother's education, wealth index, water and sanitation, and child breastfeeding are the main contributors to inequality. The share of mother's education in inequality is 31.6%; the wealth index contributes to 26.2%, whereas water and sanitation and child breastfeeding account for 17.2% and 11.2% of the total inequality.

Furthermore, birth order explains 7%, mother's working status explains 3.9%, and vaccination coverage explains 1.5%, while household size and child sex explain 0.57% and 0.48%, respectively.

Furthermore, the analysis also revealed the regional variations in inequality in Pakistan. Mother's education is the highest contributor to inequality of opportunity in stunting, accounting for 23.6% and 44.9% in rural and urban areas, respectively. The contribution of water and sanitation is 22%, the wealth index contributes 20% and child breastfeeding accounts for 15.7% of total inequality in rural areas. In urban areas, the contribution of water and sanitation to inequality is approximately 2.8%, while the wealth index contributes 32.2% and child breastfeeding accounts for 9.5% of total inequality. The share of other factors is relatively low, contributing from 0.5% to 10% in rural regions and from 0.75% to 10% in urban areas. The coverage of vaccinations accounts for 2% of inequality in rural areas while 1.7% in urban areas. Household size contributes 0.64% in rural areas and 0.80% in urban areas, whereas mothers currently working account for 5.7% and 0.24% inequality in rural and urban areas, respectively. Regarding child characteristics in rural areas, birth order contributes 9.6% to inequality, whereas child sex contributes 0.5%. In urban areas, birth order contributes marginally (0.75%) and child sex contributes 6.7% to inequality.

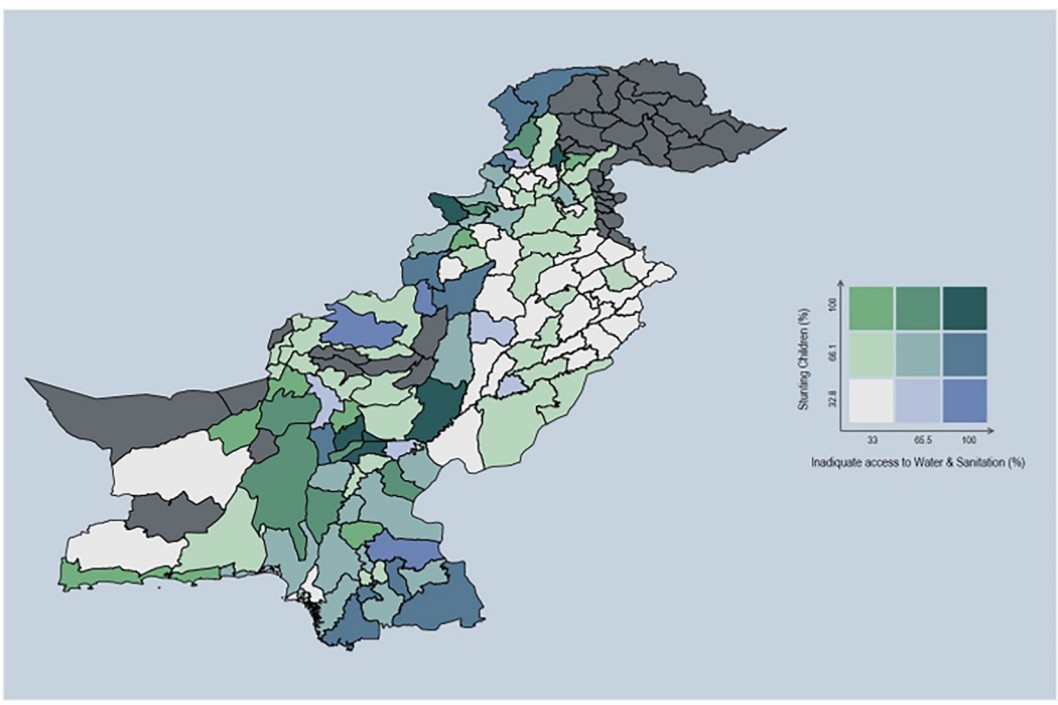

Data source: The 2017-18 Pakistan Demographic and Health Survey (PDHS)

**Fig 2. Stunting and access to water & sanitation services.**

## 4. Discussions

The findings revealed that inequality is more pronounced in urban areas than rural areas. Over the last few decades, the urban-rural disparity has widened, and it has been found that urban children face higher challenges in accessing opportunities due to multiple factors, such as high cost of living and expensive medical care [53–55]. The decomposition finding shows mother's education as an important predictor of inequality in child health, as educated mothers are more likely to possess essential knowledge about child health practices, which contributes to better child nutritional outcomes [56,57]. Moreover, education equips mothers to make informed decisions regarding various aspects of children's health and well-being, ultimately leading to improved health outcomes. The insights from these results underscore the importance of policy interventions that enhance mother's education to reduce inequality of opportunity among children [58].

Our study identified the wealth index as the second most significant predictor of child health inequality among all other factors considered for both rural and urban areas. In similar studies, the wealth index has been found to be the leading contributor to child health inequality [59,60]. Families with low socioeconomic status often encounter obstacles in providing their children with essential resources, including proper nutrition, sanitation facilities and access to healthcare. These inequalities in resources hinder children's growth and increase their risk of stunting. To effectively reduce high stunting rates in Pakistan, it is imperative to implement strategies aiming to address inequalities in opportunity.

Water and sanitation facilities are a major factor in child health inequality in rural areas, while their impact is relatively lower in urban areas. Our findings align with the studies that showed water and sanitation as dominant factors in child health inequality in rural areas [61].

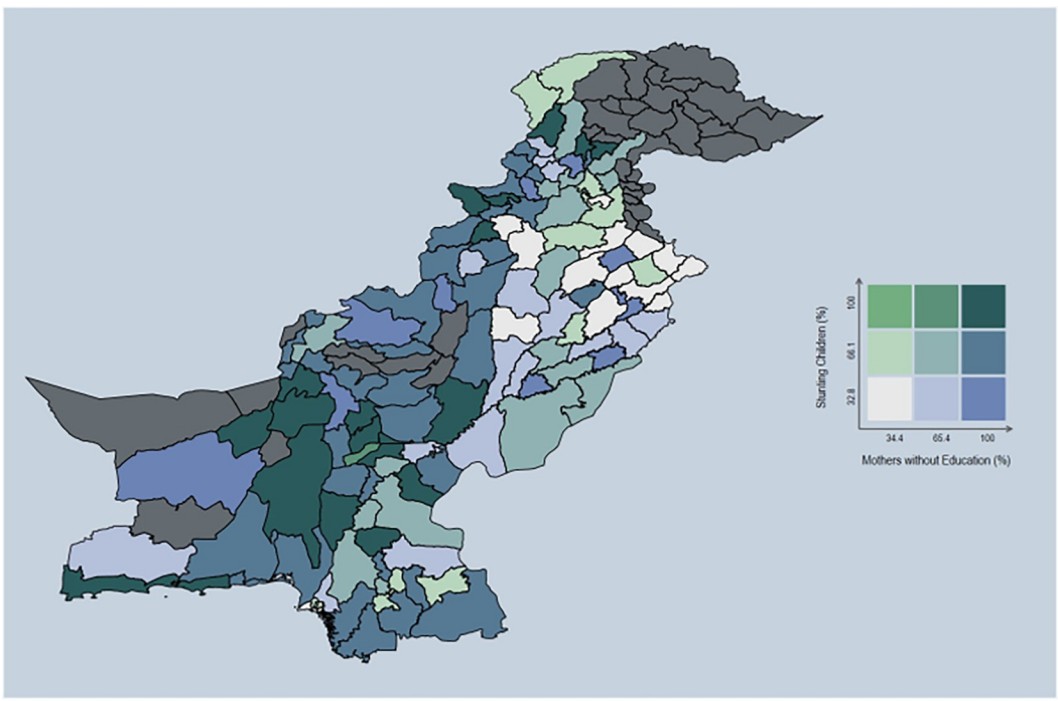

Data source: The 2017-18 Pakistan Demographic and Health Survey (PDHS)

**Fig 3. Stunting and mothers without education.**

This inequality is primarily due to limited access to water and proper sanitation facilities in rural areas, whereas urban areas have generally improved water and sanitation infrastructure. The inadequacy of water resources and improper sanitation facilities increases the risk of frequent illnesses among children, leading to malnutrition.

This study has found child breastfeeding practices to be a contributor to inequality in both rural and urban areas. In rural areas, about 16% of the total inequality is attributed to breastfeeding practices, while in urban areas, this inequality accounts for slightly lower than 10%. The present literature describes breastfeeding practice as a protective measure against stunting [62,63]. Breastmilk is a vital source for the growth and development of children; it supplies essential nutrients and helps fight against diseases [64].

Children who do not receive exclusive breastfeeding have a higher probability of experiencing stunting and wasting compared to those who are exclusively breastfed [65]. The finding shows varying effects of birth order in rural and urban areas, contributing 10% to the total inequality, while its contribution is below 1% in urban areas. This aligns with the reality that rural families tend to be larger, often resulting in unwanted births. As family size expands, children may experience a lack of attention and insufficient resources, such as food and healthcare, allocated for their well-being [66]. There is a need to raise awareness about family planning in rural areas, particularly to reduce the inequality of opportunity in child health.

Full immunization coverage and household size have been found to be minor contributors to total inequality in both rural and urban areas. The influence of vaccination coverage is less than 3% and household size contributes less than 1%. The findings do not align with the previous literature that supports vaccination coverage as a factor in reducing malnutrition [67], whereas household size and wealth were previously identified as leading contributors to child health inequalities [59,60].

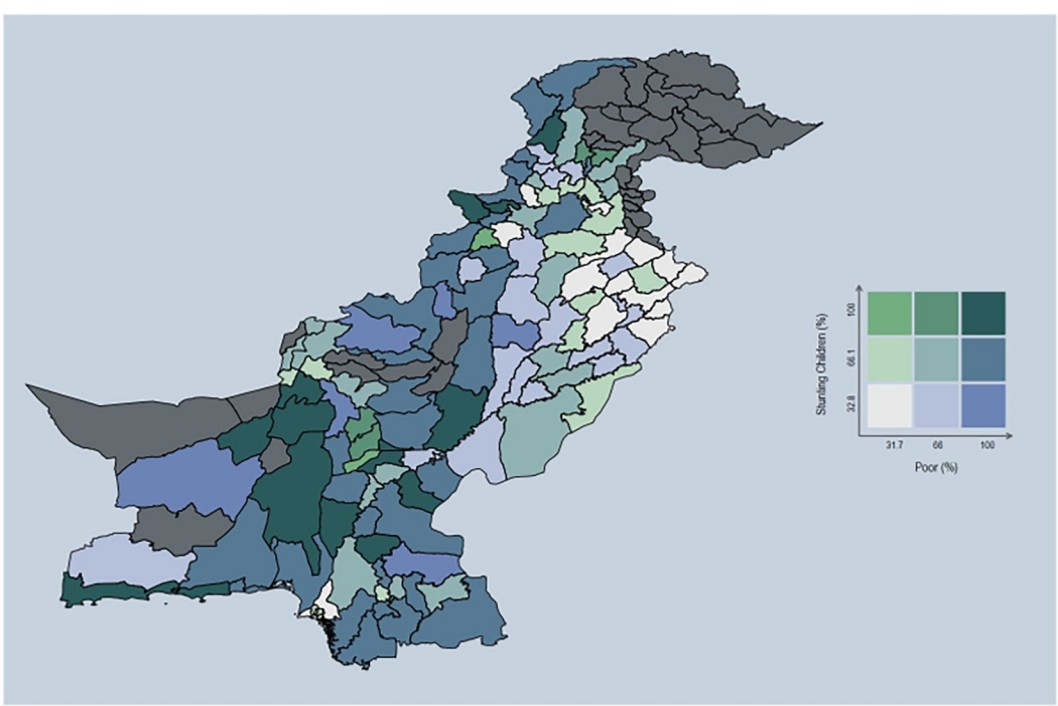

Source: The 2017-18 Pakistan Demographic and Health Survey (PDHS)

**Fig 4. Stunting and household with poor wealth index.**

This study found mother's employment status contributes 5.8% to the total inequality in rural areas, while in urban areas its contribution is below 1%. These results align with studies that demonstrate the impact of maternal employment on child health status [68]. The observed difference can be attributed to socio-economic challenges faced by rural working mothers and the absence of adequate childcare facilities. In contrast, high income levels and supportive infrastructure in urban areas facilitate better child health [69]. The analysis also showed that child sex contributes to less than 1% of total inequality in rural areas but accounts for 6.7% of total inequality in urban areas. These findings are unexpected because child sex preference tends to persist more in rural areas, where gender bias norms are strong and families are less educated.

The analysis of inequality of opportunity in childhood stunting in Pakistan underscores the need for targeted and pragmatic policy interventions that can address the structural barriers faced by disadvantaged populations. First, improving access to healthcare and nutrition services is critical to mitigating the long-term impacts of stunting. Expanding and strengthening

**Table 3. Decomposition of Inequality of opportunity in stunting (Oaxaca decomposition of D-index).**

|  | Coefficients | |
| --- | --- | --- |
| **Distribution** | **Rural** | **Urban** |
| Rural | 0.045 | 0.037 |
| Urban | 0.089 | 0.094 |
| **Method** | **Absolute** | |
| PdB (dissimilarity index) | 0.196 | |
| Ws (adapted DI) | 0.293 | |

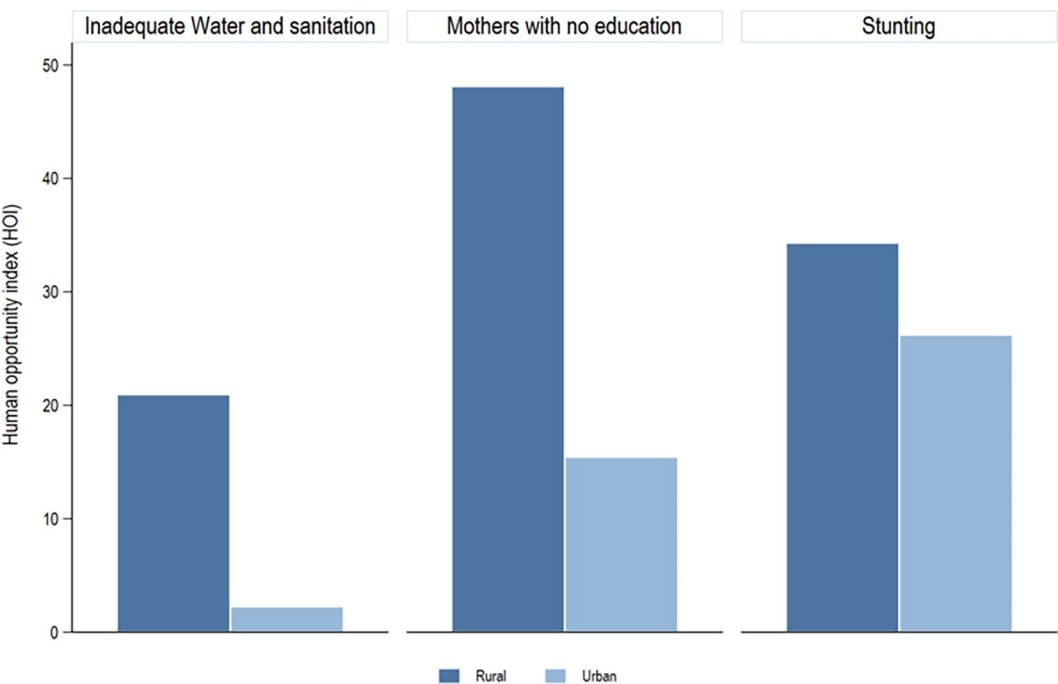

Source: The 2017-18 Pakistan Demographic and Health Survey (PDHS)

**Fig 5. Human Opportunity Index (HOI).**

community-based healthcare systems, such as the Lady Health Workers (LHW) Program, can enhance outreach to vulnerable populations, particularly in rural areas and urban slums where access to formal healthcare is limited. By increasing the capacity and reach of the LHWs, the government can ensure that essential maternal and child health services—such as antenatal care, immunizations, and nutritional counseling—are provided in underserved communities. Moreover, scaling up the Ehsaas Nashonuma Program, which focuses on providing conditional cash transfers linked to health and nutrition outcomes for children under two years of age, can further enhance early childhood nutrition and reduce stunting. This program could be expanded to include more comprehensive support, such as micronutrient supplementation and fortified foods, to address specific nutritional deficits.

**Table 4. Shapley decomposition of inequality of opportunity in stunting.**

| Variable | Full sample | | Rural | | Urban | |
|---|---|---|---|---|---|---|
| | Value | % | Value | % | Value | % |
| Coverage of full vaccinations | 0.003 | 1.51 | 0.005 | 2.03 | 0.002 | 1.77 |
| Mother's education | 0.062 | 31.68 | 0.056 | 23.68 | 0.062 | 44.96 |
| Mother currently working | 0.008 | 3.93 | 0.014 | 5.79 | 0.000 | 0.24 |
| Household size | 0.001 | 0.57 | 0.002 | 0.64 | 0.001 | 0.80 |
| Birth order | 0.014 | 7.04 | 0.023 | 9.66 | 0.001 | 0.75 |
| Child Sex | 0.001 | 0.48 | 0.001 | 0.52 | 0.009 | 6.75 |
| Wealth index | 0.051 | 26.24 | 0.047 | 20.03 | 0.044 | 32.27 |
| Water and sanitation | 0.034 | 17.26 | 0.052 | 21.93 | 0.004 | 2.84 |
| Child breastfeeding | 0.022 | 11.29 | 0.037 | 15.71 | 0.013 | 9.54 |

Second, enhancing social safety nets is essential for alleviating the economic hardships that contribute to malnutrition and inequality of opportunity. The Ehsaas Program, Pakistan's flagship social protection initiative, already provides financial support to low-income households, but its impact on child health could be amplified by integrating more nutrition-sensitive approaches. For instance, conditional cash transfers could be tied to healthcare-seeking behaviors, such as regular health checkups for children and mothers, vaccination adherence, and participation in nutrition education programs. This would not only improve healthcare utilization but also directly impact child growth outcomes. Similarly, the Benazir Income Support Programme (BISP) could be leveraged to include direct nutrition interventions, such as vouchers for nutritious foods or subsidized access to fortified food products, ensuring that the most vulnerable families can meet their children's nutritional needs.

Third, addressing the environmental factors that contribute to poor health outcomes is crucial. Improving water, sanitation, and hygiene (WASH) infrastructure is key to reducing the incidence of infections and diseases that exacerbate malnutrition. The Clean Green Pakistan Movement and the Pakistan Approach to Total Sanitation (PATS) provide important frameworks for improving sanitation and reducing open defecation, but these initiatives need to be scaled up, particularly in rural areas and informal urban settlements where poor sanitation is pervasive. Expanding access to clean drinking water and improving sanitation facilities in schools and communities would reduce the burden of waterborne diseases, such as diarrhea, which directly contribute to stunting.

Finally, the promotion of economic inclusion for marginalized segments, particularly women and low-income families, is vital for reducing inequality of opportunity. Expanding access to microfinance and vocational training through programs like the Kamyab Jawan Program can empower women to generate income and invest in their families' health and nutrition. Similarly, ensuring that poor households have access to economic opportunities through public works programs or social entrepreneurship initiatives would help them achieve economic stability, reducing the intergenerational transmission of poverty and malnutrition.

In conclusion, these policy recommendations—enhanced access to healthcare and nutrition services, strengthened social safety nets, improved WASH infrastructure, and economic inclusion—are aligned with Pakistan's current development strategies. However, to truly address inequality of opportunity in childhood stunting, these initiatives must be expanded, integrated, and targeted more effectively to reach the most vulnerable populations with a coordinated, multi-sectoral approach.

## 5. Conclusions

This study examined the inequality of opportunity in child nutritional status across regions in Pakistan, focusing on key circumstantial variables such as maternal education, family wealth status, and WASH (water, sanitation, and hygiene)—factors beyond children's control that contribute to inequality. Data from the Pakistan Demographic Health Survey (PDHS) 2017–18 was used to assess the contribution of each circumstantial variable to inequality of opportunity, applying Oaxaca-Blinder and Shapley Decomposition techniques.

The findings reveal varying contributions of the selected variables to inequality in child health between rural and urban areas. A significant portion of inequality of opportunity is attributable to maternal education and wealth index in both rural and urban areas. However, other variables, such as maternal employment status and birth order, also show notable contributions to inequality. In rural areas, water and sanitation have a higher contribution, followed by breastfeeding practices, birth order, and basic vaccination coverage. In contrast, these variables contribute less to inequality in urban areas.

The study also found that child sex significantly contributes to inequality in child nutrition in urban areas, while maternal employment status plays a more prominent role in rural areas. Household size has a minimal contribution to child health inequality in both urban and rural areas.

Furthermore, this research highlights the challenges in achieving Sustainable Development Goal (SDG) target 2.2, which aims to reduce stunting rates in Pakistan. The findings underscore the need for targeted interventions, emphasizing the importance of enhancing maternal education, improving water and sanitation infrastructure, and addressing economic disparities to reduce inequality in child nutrition.

## Author Contributions

**Conceptualization:** Shyamkumar Sriram, Lubna Naz.

**Data curation:** Shyamkumar Sriram, Lubna Naz.

**Formal analysis:** Shyamkumar Sriram, Lubna Naz.

**Investigation:** Shyamkumar Sriram.

**Methodology:** Shyamkumar Sriram, Lubna Naz.

**Project administration:** Shyamkumar Sriram.

**Resources:** Shyamkumar Sriram.

**Software:** Shyamkumar Sriram, Lubna Naz.

**Supervision:** Shyamkumar Sriram, Lubna Naz.

**Validation:** Shyamkumar Sriram.

**Visualization:** Shyamkumar Sriram.

**Writing – original draft:** Shyamkumar Sriram, Lubna Naz.

**Writing – review & editing:** Shyamkumar Sriram, Lubna Naz.

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
