## [Decision Letter · Decision Letter 0]

4 Sep 2024

PONE-D-24-10557Inequality of Opportunity in Child Nutrition in PakistanPLOS ONE

Dear Dr. Sriram,

Thank you for submitting your manuscript to PLOS ONE. After careful consideration, we feel that it has merit but does not fully meet PLOS ONE’s publication criteria as it currently stands. Therefore, we invite you to submit a revised version of the manuscript that addresses the points raised during the review process.

We look forward to receiving your revised manuscript.

Kind regards,

Jayanta Kumar Bora, PhD

Academic Editor

PLOS ONE

Journal Requirements:

2. Please ensure that you refer to Figure4 in your text as, if accepted, production will need this reference to link the reader to the figure.

3. We note that [Figures 1-3] in your submission contain [map/satellite] images which may be copyrighted. All PLOS content is published under the Creative Commons Attribution License (CC BY 4.0), which means that the manuscript, images, and Supporting Information files will be freely available online, and any third party is permitted to access, download, copy, distribute, and use these materials in any way, even commercially, with proper attribution. For these reasons, we cannot publish previously copyrighted maps or satellite images created using proprietary data, such as Google software (Google Maps, Street View, and Earth). For more information, see our copyright guidelines: http://journals.plos.org/plosone/s/licenses-and-copyright.

a. You may seek permission from the original copyright holder of Figures 1-3 to publish the content specifically under the CC BY 4.0 license.  

Reviewers' comments:

Reviewer's Responses to Questions

**Comments to the Author**

1. Is the manuscript technically sound, and do the data support the conclusions?

Reviewer #1: Yes

Reviewer #2: Yes

2. Has the statistical analysis been performed appropriately and rigorously? 

Reviewer #1: Yes

Reviewer #2: Yes

3. Have the authors made all data underlying the findings in their manuscript fully available?

Reviewer #1: Yes

Reviewer #2: Yes

4. Is the manuscript presented in an intelligible fashion and written in standard English?

Reviewer #1: Yes

Reviewer #2: Yes

5. Review Comments to the Author

Reviewer #1: The authors analyzed and presented the Inequality of Opportunity (IOP) in child nutrition in Pakistan with examining the the major determinants of inequality. The manuscript has importance in terms of uniqueness and analysis methods adopted, though the data was a bit older. The following points could be made:

1. The introduction is perfectly articulated and thus reflecting the nature of the IOP in Pakistan. However, this seems like enough data, information and literature has already been made on the topic. So, authors should provide clear research gaps and justify their current research questions to indicate the rationale of the new analysis.

2. Though the data used for this study was curated from the PDHS, however, it would be recommended to introduce more about the how data was collected, sampled and curated in for the very primary by the NIPS, and how that was curated for the current study.

3. Authors are recommended to provide justifications for the variables they used for the analysis of IOP.

4. Including some other similar, and highly relevant variables like food security, age (though it was base of of the sample selection, the age group (months), access to healthcare, would enhance the clarity of the hypothesis and study outcomes.

5. Other than 'stunting', the wasting, underweight and under-nutrient are some other indicators for measuring the IOP. But in this paper, authors only focused on stunting. Is there any justification? why not others?

6. Prevalence of IOP is apparently seen higher among the urban children, however, in general sense, the urban population are highly privileged and opportunist in terms of access to services, healthcare facilities and other economic opportunities in urban sprawl in compared to the rural. Could clearly clarify why this is the nature?

7. Would this paper suggest some pragmatic recommendations like -improving access to healthcare and nutrition services, enhancing social safety nets programs, and promotion of accessibility to the economic inclusions for the segments - based on on the study outcomes?

Thank you

Reviewer #2: The major strength of the publication is in the analysis technique employed and the introduction of spatial relationship into the analysis make the work unique. The findings show that inequality is more pronounced in the urban. The data is available in open source and can be replicate.

6. PLOS authors have the option to publish the peer review history of their article (what does this mean?). If published, this will include your full peer review and any attached files.

Reviewer #1: **Yes: **Md Al Amin

Reviewer #2: No

---

## [Author Response · Author response to Decision Letter 0]

24 Oct 2024

The introduction is perfectly articulated and thus reflecting the nature of the IOP in Pakistan. However, this seems like enough data, information and literature has already been made on the topic. So, authors should provide clear research gaps and justify their current research questions to indicate the rationale of the new analysis.

While national-level studies on child stunting are abundant, there is limited research focusing on the sub-national or provincial disparities in Pakistan. Many studies tend to aggregate the data at the national level, masking regional inequalities that may have distinct drivers and implications. Moreover, majority of previous stunting studies focus on inequality in outcomes (such as the prevalence of stunting), but relatively a few apply the inequality of opportunity framework for deeper understanding of structural inequalities, which distinguishes between factors children cannot control (e.g., parental education, socioeconomic background) and those they can (e.g., individual efforts). Further, there is a dearth of literature in unravelling the contribution of various factors in overall inequality in stunting via decomposition techniques. Additionally, very few studies in Pakistan have explored the association between access to water and sanitation and stunting. Hence, this study was undertaken to abridge the gap in the previous research using a more rigorous and nuanced approach

"The introduction part has been revised to incorporate the rationale clearly"

Though the data used for this study was curated from the PDHS, however, it would be recommended to introduce more about how data was collected, sampled and curated in for the very primary by the NIPS, and how that was curated for the current study.

"The methodology section has been revised to include the data collection and sampling strategy"

Authors are recommended to provide justifications for the variables they used for the analysis of IOP.

"The rationale of the variables used in the analysis of the IOP has been included in the revised version of the manuscript"

Including some other similar, and highly relevant variables like food security, age (though it was base of of the sample selection, the age group (months), access to healthcare, would enhance the clarity of the hypothesis and study outcomes.

The study is based on anthropometric measures which indicates nutritional insecurity among children. The age of the child and sex have been included in the analysis. The data on food security for children is not available in the PDHS, nor the study aims to look at the food security among households and link it up with nutritional insecurity among children. 

Other than 'stunting', the wasting, underweight and under-nutrient are some other indicators for measuring the IOP. But in this paper, authors only focused on stunting. Is there any justification? why not others?

Rationale of using stunting: 

Firstly, Pakistan has one of the highest stunting rates in the world. According to the Pakistan Demographic and Health Survey (PDHS) 2017-18, approximately 38% of children under five in Pakistan are stunted, compared to lower rates for wasting (7%) and underweight (23%).

Secondly, Stunting reflects chronic malnutrition, which results from long-term deprivation of adequate nutrition, healthcare, and living conditions during critical growth periods in early childhood. It signifies more persistent, structural inequalities. Unlike wasting (which reflects acute malnutrition) or underweight (which could indicate both acute and chronic malnutrition), stunting captures long-term inequality in opportunity that children experience due to socioeconomic and environmental factors. Stunting reflects structural disadvantages that accumulate over time, such as poor maternal health, poverty, and lack of access to quality healthcare and sanitation, making it more suitable for examining the deep-rooted inequalities in Pakistan.

Thirdly, In the context of IOp, stunting reflects inequalities in circumstances beyond a child’s control, such as parental education, socioeconomic status, geographic location, and access to healthcare and clean water. Because stunting is tied to long-term deprivation, it is a more appropriate marker for structural inequality compared to wasting or underweight, which may fluctuate more with temporary crises.Stunting often correlates with disparities in rural vs. urban areas, regional inequalities, and educational attainment of parents. For example, children from poorer provinces (such as Balochistan) or from lower socioeconomic backgrounds tend to exhibit higher stunting rates, making it a valuable metric for exploring IOp. Stunting is a comprehensive indicator of deep-seated inequality, capturing the multi-dimensional nature of disadvantage in Pakistan, from health and nutrition to education and socioeconomic conditions.

Fourthly, Stunting is strongly associated with impaired cognitive development, which affects a child’s educational outcomes, earning potential, and productivity later in life. Wasting, while serious, tends to have more short-term implications (i.e., increased risk of mortality due to acute malnutrition). In the context of inequality of opportunity, stunting has far-reaching impacts on future human capital development. Children who are stunted are less likely to reach their full potential, exacerbating intergenerational poverty and limiting social mobility. The long-term effects of stunting on cognitive development and economic outcomes make it a critical indicator for understanding inequality of opportunity in Pakistan. Tackling stunting addresses more than just physical health; it also has implications for educational achievement and economic inequality.

Prevalence of IOP is apparently seen higher among the urban children, however, in general sense, the urban population are highly privileged and opportunist in terms of access to services, healthcare facilities and other economic opportunities in urban sprawl in comparison to the rural. Could clearly clarify why this is nature?

The higher inequality of opportunity in child stunting among urban children in Pakistan is driven by several factors:

1. The existence of urban slums and informal settlements where living conditions are poor and access to services is extremely limited.

2. Wider socioeconomic disparities within urban areas, creating stark inequalities between the rich and the poor.

3. The vulnerability of migrant families and the lack of support systems for the urban poor.

4. Fragmented access to services and healthcare, where the poorest urban residents often face the greatest challenges in accessing quality care.

5. The high cost of living in urban areas, contributing to food insecurity and malnutrition among the urban poor.

6. Environmental hazards and poor sanitation in urban slums, leading to higher rates of disease and malnutrition.

These factors combine to make urban inequality of opportunity more pronounced, particularly for children living in poverty within urban areas, despite the general assumption that urban environments are more privileged. This paradox highlights the importance of addressing intra-urban disparities to reduce inequality of opportunity in child health outcomes like stunting.

Would this paper suggest some pragmatic recommendations like -improving access to healthcare and nutrition services, enhancing social safety nets programs, and promotion of accessibility to the economic inclusions for the segments - based on the study outcomes?

"The manuscript has been revised to include policy recommendations"

---

## [Decision Letter · Decision Letter 1]

16 Jan 2025

Inequality of Opportunity in Child Nutrition in Pakistan

PONE-D-24-10557R1

Dear Dr. Shyamkumar Sriram,

We’re pleased to inform you that your manuscript has been judged scientifically suitable for publication and will be formally accepted for publication once it meets all outstanding technical requirements and editorials comments.

Kind regards,

Jayanta Kumar Bora,PhD

Academic Editor

PLOS ONE

Additional Editor Comments

• In section 3.2 Trends in Stunting, Figure 1 shows the trend in stunting from 1990-91, 2012-13, and 2017-18 using Pakistan…. Please check the graph and give proper labeling, as you have mentioned, the trend should show the time on the X-axis.

• Please check figure 1 seems duplicated. Check the numbering of tables and figures correctly

• Overall editing and formatting are required as per the journal standard.

Reviewers' comments:

Reviewer's Responses to Questions

**Comments to the Author**

1. If the authors have adequately addressed your comments raised in a previous round of review and you feel that this manuscript is now acceptable for publication, you may indicate that here to bypass the “Comments to the Author” section, enter your conflict of interest statement in the “Confidential to Editor” section, and submit your "Accept" recommendation.

Reviewer #1: All comments have been addressed

2. Is the manuscript technically sound, and do the data support the conclusions?

Reviewer #1: Yes

3. Has the statistical analysis been performed appropriately and rigorously? 

Reviewer #1: Yes

4. Have the authors made all data underlying the findings in their manuscript fully available?

Reviewer #1: Yes

5. Is the manuscript presented in an intelligible fashion and written in standard English?

Reviewer #1: Yes

6. Review Comments to the Author

Reviewer #1: The authors have perfectly addressed all comments and suggestions. The policy recommendations are wisely articulated to link up with policy translations.

7. PLOS authors have the option to publish the peer review history of their article (what does this mean?). If published, this will include your full peer review and any attached files.

Reviewer #1: **Yes: **Md Al Amin

---

## [Editor Report · Acceptance letter]

30 Jan 2025

PONE-D-24-10557R1 

PLOS ONE

Dear Dr. Sriram, 

I'm pleased to inform you that your manuscript has been deemed suitable for publication in PLOS ONE. Congratulations! Your manuscript is now being handed over to our production team.

Kind regards, 

on behalf of

Dr. Jayanta Kumar Bora 

Academic Editor

PLOS ONE